# Attachment Trauma Is Associated with White Matter Fiber Microstructural Alterations in Adolescents with Anorexia Nervosa before and after Exposure to Psychotherapeutic and Nutritional Treatment

**DOI:** 10.3390/brainsci13050798

**Published:** 2023-05-14

**Authors:** Manuela Gander, Lukas Lenhart, Ruth Steiger, Anna Buchheim, Stephanie Mangesius, Christoph Birkl, Nina Haid-Stecher, Martin Fuchs, Anna Libal, Agnieszka Dabkowska-Mika, Elke Ruth Gizewski, Kathrin Sevecke

**Affiliations:** 1Institute of Psychology, Leopold-Franzens-University of Innsbruck, 6020 Innsbruck, Austria; 2Department of Child and Adolescent Psychiatry, Tirol Kliniken, 6060 Hall in Tirol, Austria; 3Department of Neuroradiology, Medical University of Innsbruck, 6020 Innsbruck, Austria; 4Neuroimaging Research Core Facility, Medical University of Innsbruck, 6020 Innsbruck, Austria; 5Department of Child and Adolescent Psychiatry, Medical University of Innsbruck, 6020 Innsbruck, Austria

**Keywords:** anorexia nervosa, white matter, fractional anisotropy, attachment, adolescence

## Abstract

In the present study, we explore the role of attachment for microstructural white matter (WM) changes in adolescents with anorexia nervosa (AN) before and after exposure to short-term and nutritional treatment. The case sample consisted of 22 female adolescent inpatients with AN (mean age: 15.2 ± 1.2 years) and the control sample were 18 gender-matched healthy adolescents (mean age: 16.8 ± 0.9 years). We performed a 3T MRI in the patient group during the acute state of AN and after weight restoration (duration: 2.6 ± 1 months) and compared the data to a healthy control group. To classify attachment patterns, we used the Adult Attachment Projective Picture System. In the patient sample, over 50% were classified with an attachment trauma/unresolved attachment status. Prior to treatment exposure, fractional anisotropy (FA) reductions and concordant mean diffusivity (MD) increases were evident in the fornix, the corpus callosum and WM regions of the thalamus, which normalized in the corpus callosum and the fornix post-therapy in the total patient sample (*p* < 0.002). In the acute state, patients with an attachment trauma demonstrated significant FA decreases compared to healthy controls, but no MD increases, in the corpus callosum and cingulum bilaterally, which remained decreased after therapy. Attachment patterns seem to be associated with region-specific changes of WM alterations in AN.

## 1. Introduction

Anorexia nervosa (AN) is characterized by an extremely low body weight, restriction of calorie intake, a distorted body image and fear of weight gain. This psychiatric disorder usually develops in adolescence, which is a crucial period for social, personal and biological development [1]. In this context, individuals with AN often have traumatic experiences during their childhood, such as parental divorce, mental health problems of or emotional abuse by their caregivers [2]. Thus, research has postulated that the attachment quality might have an important impact on how young people deal with the exposure to these adverse life events [3]. However, the impacts of attachment quality on the underlying pathophysiology remain largely unknown [4], particularly in the field of white matter (WM) microstructure associated with AN [5]. 

In recent years, numerous studies have used diffusion tensor imaging (DTI) to estimate WM integrity within the brain. In WM, the Brownian motion of water molecules is restricted by the microstructure of WM fiber bundles, resulting in an anisotropic diffusion. Two key measures of DTI to describe the microstructural properties of WM are the fractional anisotropy (FA) and mean diffusivity (MD). FA is a dimensionless measure normalized between 0 and 1 which describes the anisotropy of the diffusion within a voxel, where 0 represents fully isotropic diffusion and 1 represents fully anisotropic diffusion. MD reflects the average diffusion within brain tissue irrespective of direction. FA is sensitive to neural tissue integrity [6,7,8], whereas MD is related to extracellular volume and inflammation. Investigating cortical networks and circuitries remains crucial as brain regions are interconnected via axons which are considered to modulate human behaviors, cognitions, emotions and other clinical features [9]. A manageable amount of research applying cross-sectional study designs has used DTI to assess microstructural integrity in AN patients [10,11]. Compared to healthy individuals, decreased FA values were reported in WM tracts of the cingulum, fronto-occipital and fimbria-fornix circuits [12,13]. These results have been relatively inconsistent and may arise from differences in subject characteristics (e.g., duration of illness), methodological approaches and neurodevelopmental factors (e.g., age) [12,14,15,16,17,18,19,20,21]. Among previous research groups that longitudinally assessed axonal integrity in acutely ill AN patients, altered FA were most consistently found in WM tracts of the cingulate gyrus, the corpus callosum and the fornix, as well as the superior longitudinal fasciculus [13,22], which restored after therapy to varying degrees [5,22,23], or remained impaired [24,25,26]. Alterations in MD measures are typically negatively correlated with FA [27,28] and thought to be suggestive of inflammation or atrophy by representing water content or extracellular fluid [29]. 

As some researchers found a complete reversal of WM alterations after weight recovery, it was hypothesized that the observed structural brain alterations are more likely to represent consequences of malnutrition (state-related) than preexisting risk factors (traits) [5,23]. On the other hand, some studies could not replicate these findings. Their results show associations of WM alterations with AN-related characteristics, such as symptom severity, harm avoidance and anxiety [16,30]. This raises important questions of whether changes of the WM microstructure during treatment are primarily state-related due to undernutrition or whether they are more enduring abnormalities resulting from certain predisposing factors [5]. 

Attachment might represent such a predisposing factor that possibly impacts the WM microstructure in AN. A secure attachment pattern is associated with greater resilience when confronted with stress or trauma [31]. Caregivers of secure individuals are experienced as available and sensitive towards their needs. Consequently, secure individuals can reach out for comfort and safety when confronted with stressors and they show adaptive coping strategies and skills in their conflict management [31,32]. On the other hand, individuals classified with an insecure attachment status experienced ruptures in their attachment relationships. In case of an insecure-dismissing representation, caregiver’s often respond with rejection to their attachment cues. Thus, these individuals use deactivating strategies to maintain a distance in attachment relationships. The insecure-preoccupied attachment status is characterized by experiences of caregivers’ inconsistent responses to their attachment needs. As a strategy to prevent the breakthrough of traumatic attachment-related memories, these individuals primarily use cognitive disconnection defensive strategies to confuse or obscure their attachment relationships. Secure, insecure-dismissing and insecure-preoccupied patterns are defined as resolved patterns because they allow the individual to stay organized when dealing with stressful attachment situations [31]. 

Unresolved/disorganized individuals, however, do not find comfort and solace when facing threatening events. Consequently, they become overwhelmed by experiences of severe distress, which leaves them in a state of helplessness and dysregulation [33]. This attachment status is rooted in past experiences of threatened abandonment (i.e., parental suicide or loss) or danger by their attachment figure (i.e., sexual abuse) [34,35]. These severe disruptions in the attachment relationship put individuals at risk for developing attachment trauma [36]. Attachment trauma is considered as a form of relational trauma and its presence can be assessed using an attachment interview (i.e., the Adult Attachment Projective Picture System (AAP) or the Adult Attachment Interview (AAI)) [37]. 

A growing body of research demonstrates that unresolved attachment patterns are related to a high risk for the onset of mental disorders and a higher symptom severity [32]. Furthermore, attachment disorganization can have a negative influence on the psychotherapeutic process and the outcome [38]. Thus, establishing a secure base allowing patients to explore their self and their relationships more accurately and safely represents one of the main goals of psychotherapy. In addition, therapists can represent a temporary safe attachment figure for the patient that he or she can seek out for help and protection in times of severe distress [38]. In patients with AN, this might be particularly important as they often demonstrate impairments when coping with negative feelings [39]. Furthermore, they often show difficulties to rely on their caregivers for safety and comfort [40]. Several authors discussed that their preoccupation with their body and their excessive control of their eating behaviors might replace their needs for a secure base and gives them a false sense of control over their lives [41,42]. In interpersonal relationships, they often show underdeveloped capacities for interpersonal emotion regulation, and they are unable to find comfort in attachment figures when confronted with traumatic situations. This often results in being overwhelmed by fear and threat in these situations [33]. For psychotherapeutic treatment, these patients might be particularly challenging because they experience their therapist as a source of danger and, thus, they more often drop out of therapy or demonstrate adverse treatment outcomes [41]. Based on these research findings, it is not surprising that AN patients often show difficulties in responding to interventions that do not focus on interpersonal issues [38].

There are a handful of studies demonstrating links between a secure attachment quality and increased FA in WM regions that are associated with interpersonal competence [43,44]. However, only one study to date has focused on WM abnormalities in young patients with an unresolved attachment pattern. They found that the unresolved category was related to reduced FA values in the inferior fronto-occipital longitudinal fasciculus and the splenium of the corpus callosum [45]. Although some research groups have previously demonstrated WM microstructure changes from pre- to post-therapy [5,22,23], research on underlying psychological factors associated with these changes has only recently begun. Whereas psychological components such as symptoms of anxiety or depression were not related to gray (GM) and white matter changes in AN patients [5], Lenhart et al. [4] demonstrated in a recently published study that attachment trauma is associated with GM alterations in adolescent AN. Considering that WM alterations follow a similar pattern [5], attachment might also play a key role for changes from pre- to post-treatment. Therefore, we hypothesized that attachment trauma might also be related to WM alterations in AN patients, suggesting that they might be less responsive to traditional psychotherapeutic and nutritional treatments. 

Current literature has shown that in patients with AN, changes in the integrity of WM fibers occur in the acute stage and at least partially recover after therapy. However, there are a lack of longitudinal analyses in young age groups (i.e., 14–18 years) and of studies that have examined psychological components such as attachment status, which was associated with treatment response in AN patients [4]. 

In the first part of this work, we aim to determine whether there are differences in the FA of WM fibers in the acute disease state, whether they normalize after inpatient psychiatric treatment and weight restoration (body mass index (BMI)-for-age ≥ 5th percentile), and further, to what extent they agree with the existing literature. Since a recently published study already suggested that the attachment status of a patient is related to the therapeutic response in extensive brain regions of the GM [4], the hypothesis of the second part of the study was that patients with an attachment trauma show persisting WM fiber microstructural alterations as assessed by DTI, as attachment dysregulation under conditions of extreme stress is thought to manifest itself in specific brain structures associated with deficits in emotion regulation [46,47].

## 2. Materials and Methods

### 2.1. Participants

Our case population consisted of 27 female adolescent inpatients from the Department of Child and Adolescent Psychiatry with diagnosed anorexia nervosa (ICD-10 F50.0) and with or without attachment trauma. We recruited a gender-matched control population of 23 non-clinical adolescents from the community with no present or history of anorexia nervosa (ICD-10 F50.0) and no attachment trauma. All participants were aged between 14 and 18 years. We recruited the study sample in different areas of Austria and Southern Germany. The recruitment took place between January 2015 and June 2018. The cases and controls were ascertained by using the ICD-10 F50.0 criteria for anorexia nervosa restrictive type (present/absent). Diagnoses were based on the Structured Clinical Interview for DSM-IV (SCID-I [48]). Medical and psychological exclusion criteria were evaluated before the MR scanning procedure. The control group was assessed with the SCID-I to exclude all participants with a history of or a present psychiatric disorder (i.e., eating disorders, anxiety disorders, major depression). Multiple laboratory analyses were run in both groups. Furthermore, our clinic performed a standard medical examination to exclude patients and controls with medical conditions or metabolic diseases (i.e., chronic or acute somatic or functional diseases such as tumors, heart conditions, strokes, a history of fainting or head trauma). Additionally, we excluded patients who were extremely underweight and required pediatric treatment for medical stability and an improvement of cognitive functioning prior to their inpatient psychiatric treatment. In the patient group, all adolescent females experienced amenorrhea. In total, we excluded *n* = 5 patients due to the loss of neuropsychological follow-up (*n* = 3) and movement artifacts during the scanning procedure (*n* = 2). In the control group, we excluded *n* = 5 adolescents due to incidental findings on the MRI scan (hydrocephalus and multiple sclerosis, *n* = 2), artifacts due to movement during the MRI procedure (*n* = 1), dental brace (*n* = 1) and a history of an eating disorder in the past as assessed with the clinical interview (*n* = 1). Furthermore, we assessed all participants with the Hamburg Wechsler Intelligence Scale [49] to exclude participants with an intelligence score < 85. Further exclusion criteria were an insufficient knowledge of the German language to understand and answer the psychological questionnaires, a history of substance abuse and contraindications for an fMRI scan. Our final study sample included 22 female patients (mean age: 15.2 ± 1.2 years) and 18 female controls (mean age: 16.8 ± 0.9 years). There were no missing data in the final sample.

We recruited the patients within the first week of their admission to our specialized unit for adolescent eating disorders at the Department of Child and Adolescent Psychiatry. The case sample was exposed to psychotherapeutic treatment that included individual psychotherapeutic interventions, psychoeducation and family-based treatment. Furthermore, they were medically supervised, including a close monitoring of altered vital signs as well as abnormal laboratory findings (i.e., electrolyte or electrocardiogram changes). For each patient, an individual refeeding plan was created depending on markers of medical stability. This plan varied from a total calorie intake of 2400 to 3000 kcal per day. We reviewed a patient’s meal intake and mental condition on a daily basis. Furthermore, patients attended a hospital-based school and regular psychotherapeutic and adolescent group sessions. During their stay at the hospital, patients did not receive an attachment-based treatment. 

The patient group received their first MRI brain scan (timepoint 1 (Tp1), *n* = 22) after they reached medical stability. At Tp1, they also completed the psychological questionnaires and clinical interviews. The second MRI scan (timepoint 2 (Tp2), *n* = 18) was conducted after their weight restoration (BMI-for-age ≥ 5th percentile). In the ICD-11, AN is characterized by a significantly low body weight for the person’s height, developmental stage and age that is not due to the unavailability of food or other medical conditions. In adults, the diagnosis of AN depends on a BMI < 18.5 kg/m², as defined by the Centers for Disease Control and Prevention (CDC) and the World Health Organization (WHO). A BMI-for-age ≤ 5th percentile is commonly used for children and adolescents [50]. For our study, a stable weight recovery was therefore defined by reaching a BMI-for-age ≤ 5th percentile.

This took place on average 2.6 ± 1 months following the first session. The community-based control group underwent one MRI scan. MRI scans at Tp1 and Tp2, inpatient treatment and data collection were carried out between February 2015 and September 2018. We present our research question in a population, exposure, comparison and outcome format (PECO) in Table 1. The present study has a pre–post-intervention design with attachment trauma (classified by the AAP) as the environmental exposure and nutritional and psychotherapeutic treatment as the clinical exposure. The outcome variable was white matter microstructural alterations assessed by DTI. In addition, WM alterations were compared between patients and controls (Tp1) and between patients with attachment trauma and patients with no attachment trauma (Tp1 and Tp2). 

The study received ethical approval from the local Ethics Committee (AN2015-0036) and was carried out according to the Declaration of Helsinki. We received informed consent from all participants and their parents/legal guardians prior to the study procedure. 

### 2.2. Measures

#### 2.2.1. Eating Disorder Diagnosis and Symptomatology

We conducted the Structured Clinical Interview for DSM-IV (SCID-I, German translation [48]) to the diagnose AN restrictive type. The interview was administered by trained clinical psychologists at the Department of Child and Adolescent Psychiatry. The SCID-I is often referred to as the gold standard in determining the accuracy of DSM mental disorder diagnoses in adults and adolescents [48,51].

Furthermore, we assessed eating disorder symptom severity in patients and controls using the German version of the Eating Disorder Inventory-2 (EDI-2) [52]. The EDI-2 questionnaire consists of 91 items that are rated on a 6-point Likert scale. Its 11 subscales are: maturity fears, bulimia, body dissatisfaction, drive for thinness, ineffectiveness, perfectionism, interpersonal distrust, interoceptive awareness and three provisional subscales (asceticism, impulse regulation, social insecurity). The subscales can be added up to a total score indicating the severity of eating disorder symptoms.

#### 2.2.2. Attachment Classification

To assess adolescent attachment status, we used the Adult Attachment Projective Picture System [31]. This attachment interview consists of a set of eight picture stimuli (one neutral and seven attachment scenes) which depict attachment-related themes such as death, separation, illness and solitude. Each participant is asked to tell a hypothetical story about what is going on in the picture, what led up to that scene, what the characters are thinking or feeling and what might happen next. These series of standardized questions are asked for each picture stimuli. The attachment narratives are then coded by a reliable and certified AAP coder into four attachment categories according to the AAP manual [31]: The secure attachment pattern, which is characterized by mutual enjoyment in relationships and thoughtful self-exploration.The insecure-dismissing pattern, which is characterized by individuals using primarily deactivating elements (i.e., authoritarian orientation, distance, normalization) to create a distance to attachment relationships.The insecure-preoccupied pattern, which is characterized by narratives with a lot of confusing material and negative emotions such as anger, guilt and shame.The unresolved/disorganized attachment or attachment trauma, which refers to narratives characterized by an inability to protect oneself or seek protection or care from attachment figures when confronted with traumatic attachment-related themes such as death, abuse, emptiness or isolation [3].

Based on several empirical attachment studies [34,53,54], we used the two-group classification of unresolved and resolved (secure, insecure-dismissing, insecure-preoccupied) attachment patterns. The AAP interviews were coded by two certified and reliable AAP judges (A.B. and M.G.), who participated in an 8-day AAP workshop and reliably coded 80% on a minimum of 30 cases (for more information on certification criteria, see George and West [37]). One of the judges was a member of the International AAP Training Consortium and has led several neurophysiological and clinical research studies since 2002. Inter-rater reliability for this study demonstrated an agreement for the two-group classification (resolved–unresolved), which was 97.5%, κ = 0.947, with a narrow 95% confidence interval [0.845, 1.049], *p* < 0.001. The independent judges agreed in 39 out of 40 cases and disagreements were resolved by conference. 

#### 2.2.3. MRI Acquisition and Processing

MRI measurements were performed on a 3.0 Tesla whole-body MR scanner (Magnetom Verio, Siemens Erlangen, Germany) using a 12-channel head coil at the Department of Neuroradiology at the Medical University of Innsbruck. All participants underwent the same MRI protocol, including whole-brain T1-weighted, DTI, fluid-attenuated inversion recovery and T2- and proton density-weighted sequences. DTI was acquired using spin-echo echo-planar imaging with the following settings: echo time = 95 ms, repetition time (TR) = 7500 ms, bandwidth = 1596 Hz/pixel, matrix size 230 × 230, 45 axial slices, voxel size 2 × 2 × 2 mm^3^, 20 diffusion gradient directions, b-value of 1000 s/mm^2^ and one reference image with b = 0 s/mm^2^. These sequences were assessed by experienced neuroradiologists to exclude abnormal subclinical findings, such as white matter lesions, as structural anomalies or infarctions. 

DTI data and previously co-registered T1-weighted images were normalized onto the T1 template in Montreal Neurological Institute MNI space, and the resulting transformation parameters were applied to the participants’ corresponding FA and MD maps. Spatially normalized parametric images were smoothed with a Gaussian kernel of 8 × 8 × 8 mm^3^. An analysis-specific image mask of the standardized WM compartment and a masking threshold of 10% were applied to reduce the signal-to-noise ratio and to exclude regions not containing enough WM. MRI data were visually inspected for obvious artifacts arising from motion or instrumental failure or misalignments of brain structures. Whole-brain analysis was conducted using SPM12 (Statistical Parametric Mapping, Institute of Neurology, London, UK) while running MATLAB 9.5 (R2018b; MathWorks, Natick, MA, USA). To pre-process and analyze DTI data, statistical parametric mapping (SPM, Wellcome Department of Cognitive Neurology, London, UK) was used. The software package SPM12 implemented in MATLAB 9.5 (R2018b; MathWorks, Natick, MA, USA) objectively localizes focal DTI differences throughout the whole brain at the voxel level [55]. One advantage of the used approach is that there is no prior selection or segmentation of brain regions needed. This allows to investigate alterations in MRI parameters across the whole brain.

### 2.3. Statistical Analysis

For statistical analyses of imaging data, a flexible factorial design was set-up to compare cross-sectional and longitudinal DTI data at baseline and follow-up timepoints. A parametric model was assumed for each voxel cluster, which applied a general linear model to describe the data variability. Based on the number of resels in the image and the Random Field Theory, adjustments were made to correct for spatial correlation and multiple comparisons. 

This was followed by correction for multiple comparisons via the family-wise error rate (FWE) at the *p*-level < 0.05 to adjust for type I errors (false positives) because numerous statistical tests were conducted. Total intracranial volume and age were used as nuisance variables in all analyses to correct for different brain sizes. Additionally, ventricular size was included as a covariate in our general linear model to test the potential influence of ventricular size on DTI metrics [56].

In the second part of the study, BMI was also included as a covariate in statistical models to minimize the effect of weight- and nutrition-associated effects, as the goal of this part was to detect attachment status-related differences in DTI metrics.

Conclusions were made at *p* < 0.001 for the entire cohort (1st part) and at *p* < 0.01 for subgroups (2nd part). This was followed by correction for multiple comparisons via the family-related error rate (FWE) at the *p*-level < 0.05.

## 3. Results

### 3.1. Sociodemographic and Clinical Values

We summarize the demographic and clinical characteristics in Table 2 and Table 3. Our results demonstrate that the patient group scored higher on the EDI-2 (*p* < 0.001), and they had a significantly lower BMI (*p* < 0.001) compared to the control group. Furthermore, patients with an unresolved attachment status did not differ from patients with a resolved attachment pattern on symptom severity, as assessed with the EDI-2 and the BMI. However, the patient group had a lower BMI (*p* < 0.001) at baseline and the follow-up timepoint (*p* < 0.001) than the control group. Concerning sociodemographic characteristics, the patient group did not differ from the control group in the number of siblings, occupation, the marital status of their parents and attachment patterns (see Table 3).

### 3.2. Cross-Sectional Comparisons of FA 

At Tp1, SPM localized significant abnormalities of confluent voxel clusters of the corpus callosum and the cingulum bilaterally spreading to the fornix and the cortical tract on both sides (*p* < 0.001). Additionally, significantly lower FA was evident in areas of the inferior occipitofrontal and longitudinal fasciculus on the left hemisphere (*p* = 0.029). Ventricular size was negatively correlated with FA values predominantly of the fornix and adjacent periventricular regions (*p* < 0.001), and the inferior occipitofrontal and longitudinal fasciculus on the left side (*p* = 0.003). No WM region showed higher FA in AN patients compared to HC. In contrast, at Tp2, no significant differences between AN patients and HC were evident.

### 3.3. Longitudinal Comparisons of FA 

Within-group comparisons from Tp1 to Tp2 showed relative FA increases of WM tracts of the corpus callosum and the cingulum bilaterally, extending to the fornix (*p* < 0.001), as well as in a cluster of the inferior occipitofrontal and longitudinal fasciculus on the left side (*p* = 0.008). Ongoing FA decreases were revealed in a cluster of the right corticospinal WM tract (*p* < 0.001) (see Table 4, and Figure 1).

### 3.4. Cross-Sectional and Longitudinal Analyses of MD

MD was significantly higher in acute AN patients at Tp1 relative to HC in a broad region of the corpus callosum and the cingulum, extending to the corticospinal tract and the fornix (*p* < 0.001), the arcuate posterior segment bilaterally (*p* = 0.001) and global subcortical regions (*p* = 0.001). Longitudinal analyses revealed significant decreases in these mentioned regions from Tp1 to Tp2 (*p* < 0.001). No WM region showed decreased MD in AN patients compared to HC at Tp2 (Appendix A).

### 3.5. Cross-Sectional Comparisons of Patients with Attachment Trauma

We conducted exploratory analyses to test for possible associations with attachment trauma. In acute AN patients with attachment trauma, compared to HC, lower FA was evident in the corpus callosum and the cingulum bilaterally (*p* < 0.001). These regions were still decreased after weight restoration and psychotherapy in the left (*p* = 0.005) and in the right hemispheres (*p* = 0.007). No significant differences were found between AN patients without attachment trauma and HC at both timepoints. Further, no associations between ventricular size and DTI metrics were evident. No significant differences regarding attachment trauma were found for MD data (see Table 5, Figure 2). 

## 4. Discussion

Despite the growing number of studies exploring WM alterations in AN, the roles of psychological components for WM changes are not well-understood, particularly in adolescent age groups. The findings presented here were obtained from an adolescent inpatient sample with AN and suggest that attachment patterns are associated with region-specific microstructural integrity before and after weight restoration. Before treatment, our study sample demonstrated DTI alterations in the corpus callosum and the cingulum, bilaterally extending to the corticospinal tract and the fornix, as well as FA reductions of the left inferior occipitofrontal and longitudinal fasciculus. After the treatment, DTI metrics of these regions rapidly normalized, which is in line with some previous findings [13,57,58]. Interestingly, when focusing on attachment patterns, patients with an unresolved attachment status showed significantly lower FA in the area of the corpus callosum and posterior cingulate cortices compared to HC, which were still evident after therapy. Despite some noteworthy limitations, the findings of the present study add some clarity to the inconsistent results reported in other longitudinal studies on WM microstructure in patients with AN.

The overall pattern of WM microstructure in underweight patients prior to treatment is largely consistent with previous studies that report significantly lower FA in the corpus callosum and the fornix [30] in underweight AN relative to HC [5,22,23,59]. Furthermore, two recently published meta-analyses on whole-brain DTI studies demonstrated the largest cluster of lower FA in the corpus callosum in AN patients, followed by other subcortical areas [13,60]. The advancement of non-invasive diffusion MRI tractography techniques has enabled virtual 3D visualization of WM. The corpus callosum is the streamlined major fiber bundle facilitating the communication between the left and the right hemispheres. Crossing fibers in the anterior third of the corpus callosum cross to the forceps minor, connecting prefrontal cortices and supplementary motor areas, which are important for body image perception and task switching [61]. Corpus fibers pass transversely to the cerebral cortex to form the corona radiata, whereas rostral fibers connect the orbital regions of the frontal lobes [62]. The fornix is part of the limbic system and is closely linked to areas such as the corpus callosum and the cingulate cortex, the hypothalamus, amygdala, ventral striatum, as well as prefrontal and orbitofrontal cortices [63].

The fornix is an area which is associated with reward-regulating behaviors. Researchers have postulated that individuals with AN often show decision-making deficits which cause an under-response to conventionally rewarding experiences, such as eating or social interaction [30]. However, much more attention has been paid to WM anomalies in the corpus callosum in patients with AN [23]. An altered WM microstructure in the corpus callosum during the acute state of AN is commonly related to altered taste processing, a selective attention to disorder-salient stimuli [15] and a distorted body image perception in AN [23]. Furthermore, the corpus callosum is involved in accurately identifying emotional states and interpreting behavior [59,64]. Several authors report that majority of patients with AN show high levels of alexithymia, which is characterized by difficulties in identifying, describing and communicating feelings and deficits in differentiating between bodily sensations and feelings [65]. It has been hypothesized that the corpus callosum plays a critical role for alexithymia symptoms [64,66], and thus it is not surprising that lower FA in this region has been one of the most consistent findings in patients with AN during the acute phase of their illness [5,23].

In light of our longitudinal observations of increases in the fornix and corpus callosum after weight restoration in the total sample, it seems likely that the WM microstructure partially normalizes with weight gain. However, findings of longitudinal studies are inconsistent. While some researchers report an increase in FA in certain brain regions after treatment [5,22], others could not replicate these results in patients following weight restoration [67]. We found ongoing FA decreases in AN patients in corticospinal and fronto-occipital WM tracts bilaterally, which is in line with other research groups [5] that demonstrated reduced FA in the right corticospinal tract. One possible explanation is that excessive physical exercise—a common feature of AN—might be related to the observed reductions in WM. However, previous studies could not find a significant correlation between the amount of physical exercise and FA values in certain brain regions [5]. 

Therefore, researchers called for a more systematic investigation of these alterations that considers psychological components which potentially influence WM changes before and after treatment [5]. In this regard, our study demonstrated for the first time that attachment patterns might be associated with the normalization of WM after weight restoration therapy. While patients without attachment trauma did not show differences in the WM microstructure compared to the HC group after treatment, those with an attachment trauma demonstrated significant FA decreases in regions of the posterior corpus callosum bilaterally. Attachment theory represents a good framework to explain this finding. 

Theorists propose that the restrictive food intake in patients with AN might serve as a way to cope with their inner feelings [39,41]. Children who grow up in attachment relationships that are characterized by severe neglect or abuse by caregivers learn that their emotions are viewed as inacceptable or frightening, resulting in a belief that their feelings and needs must not be experienced and expressed to others [39,65]. Thus, they often feel helpless and isolated when confronted with stressful situations as they cannot reach out to their attachment figures for safety and comfort [41]. This provides a fertile ground for engaging in disordered eating behaviors in an attempt to avoid or cope with their negative emotions. Their restrictive calorie intake might provide them with a false sense of control of their lives and replace their secure base needs [42]. As the corpus callosum plays an important role in the cognitive processing of emotions [23,65,66], the reduction of WM integrity in the corpus callosum in our patient cohort with attachment trauma might reflect a consequence of these impairments that are not reversed after weight restoration therapy. This theory is further supported by findings on reduced WM integrity in the corpus callosum in young patients with an unresolved attachment status [45] and experiences of childhood maltreatment [68]. 

Furthermore, research studies suggest an impact of childhood trauma on biological traits, such as a heightened sensitivity to salient stimuli that results in inappropriate stress responses. For example, the study of Meneguzzo et al. [69] tested 24 h urinary free cortisol levels during the acute phase in patients with eating disorders (AN or bulimia nervosa). They found a reduced daily excretion of cortisol in patients with a history of childhood maltreatment compared to those with no maltreatment experiences. In line with the observed impaired functioning of the HPA axis found in the Meneguzzo et al. study [69], our results might add further data supporting the hypothesis of the presence of an ecophenotype of maltreated AN. Future research incorporating this phenotype into the evaluation of personalized treatment approaches might advance our understanding of the clinical presentation, outcome and treatment efficacy in the field of AN.

Concerning methodology, a recent meta-analysis reported similar data to our study and called for an evaluation of free-water correction in the analysis [70]. In an analysis focusing on the fornix, Kaufmann et al. [56] demonstrated that DTI metrics may be biased by CSF-induced partial volume effects (PVE) due to the ventricular enlargement typically found in acute AN, suggesting that the fornix and other periventricular regions might be affected by PVE. We addressed this issue by including ventricular size as a covariate in our general linear model. Consistent with previous studies (Ref. [5]), ventricular size was negatively correlated with FA values predominantly of the fornix, but other periventricular regions were also significantly associated with FA alterations in acute AN. Small parts of the corpus callosum might also be affected by PVE, although the changes were not extensive. Results of attachment-related analyses were not influenced by adding ventricular size as a covariate in these analyses. Due to the extensive shrinkage of GM and WM in acute AN, the introduction of a covariate such as BMI, which was already included as a covariate in the attachment-related analyses, may also partially correct for free-water itself. Additionally, the lack of evaluation of free-water correction might be related to a reduction in effect size, so our findings might underestimate the observed association. Nevertheless, free-water correction may be a potentially interesting factor for future studies.

The findings of the present study must be interpreted in the light of some limitations. First, our cohort consisted of adolescent patients ≤ 18 years, and therefore, the region-specific alterations may not be generalized to older patients with longer disease durations. In this context, the HC group was older than the patient group. Even though we considered age as a covariate, this aspect can be seen as a further limitation of this study. Our study compared the groups with ventricular size and BMI as further nuisance covariates. Future research applying other variables such as differing lengths of treatment periods, or the rate of weight gain, might expand our understanding of what other factors might account for the observed differences in the WM microstructure. In particular, comorbid psychiatric disorders such as affective disorders [57] could potentially influence WM changes and should, therefore, be considered as important covariates in future research studies. 

Second, our female patients were hospitalized and demonstrated severe AN symptomatology. Future research on out-patients with mild or moderate levels of symptoms might show different dynamics of microstructural WM brain changes. In this regard, replication studies that differentiate between in- and out-patient samples and between eating disorder subgroups may facilitate further insights into brain alterations before and after treatment. Third, our study explored patients during their hospitalization, and we did not conduct data at additional follow-up timepoints to examine their long-term recovery. Fourth, our study did not examine changes of the attachment status during the treatment and its impact on changes in the WM microstructure. Even though attachment patterns are unlikely to change over the course of a short-term treatment, longitudinal studies with several timepoints could help to clarify whether these WM abnormalities persist over a longer period. 

Furthermore, we conducted a cross-sectional analysis between the clinical and the control groups, which is in line with other research studies [5,22]. In this regard, the time period between the two MRI scans in the clinical group was relatively short (2–3 months) so we would not expect significant age-related changes. Nevertheless, some studies demonstrated that regional FA values undergo continuous, non-linear changes over the lifespan [71]. Thus, future studies should compare longitudinal data between patients and controls with longer time periods between the MRI scanning procedure to replicate these findings. 

Furthermore, we exclusively focused on attachment trauma, and we did not consider other forms of trauma (i.e., exposure to or being the witness of terrible events such as war, serious accidents, natural disasters, etc.) in our study. We did not exclude participants who suffered from another form of trauma as patients with attachment trauma can also have experienced these. Research has even demonstrated associations between attachment trauma and a higher level of PTSD symptoms after traumatic events [72]. Thus, more studies with larger sample sizes are needed to explore WM abnormalities in individuals with other forms of trauma. Fifth, although our sample size is comparable to those in previous brain imaging studies in AN [67], our results need to be replicated in large-scale research. 

Despite these limitations, the present study has a number of strengths as it circumvents some aspects that might be responsible for the inconsistent results reported in previous longitudinal studies on WM microstructure in patients with AN. On a neurodevelopmental level, our research focused exclusively on an adolescent age group between 14 and 18 years of age. As our population is of a young age, we do not have any chronic forms of AN with several years of illness in the past that might demonstrate a poorer therapeutic response and thus show different WM alterations after treatment. Methodologically, we included attachment trauma as a component that might be important to explain the observed WM alterations pre- and post-therapy. In addition, our drop-out rate at Tp2 was relatively small, so our sample size for a longitudinal design was relatively high compared to other studies (i.e., [19,22]).

## 5. Conclusions

Taken together, our findings indicate that severe attachment-related impairments also become evident in region-specific WM alterations that are related to body dissatisfaction and the cognitive processing of emotions. In other words, the decreased FA in the corpus callosum in patients with attachment trauma might be an indicator of persistent emotion dysregulation after weight recovery. For clinical practice, our results suggest that adolescent patients with AN might particularly benefit from attachment-based treatments [73], that place an emphasis on identifying and describing emotions in attachment relationships in addition to symptom reduction [65]. A more intensive intervention focusing on emotion regulation in interpersonal relationships might potentially foster efficacious change mechanisms that support AN patients to gain a secure attachment quality [38] and cause a reversal of abnormalities in their WM microstructure. 

## Figures and Tables

**Figure 1 brainsci-13-00798-f001:**
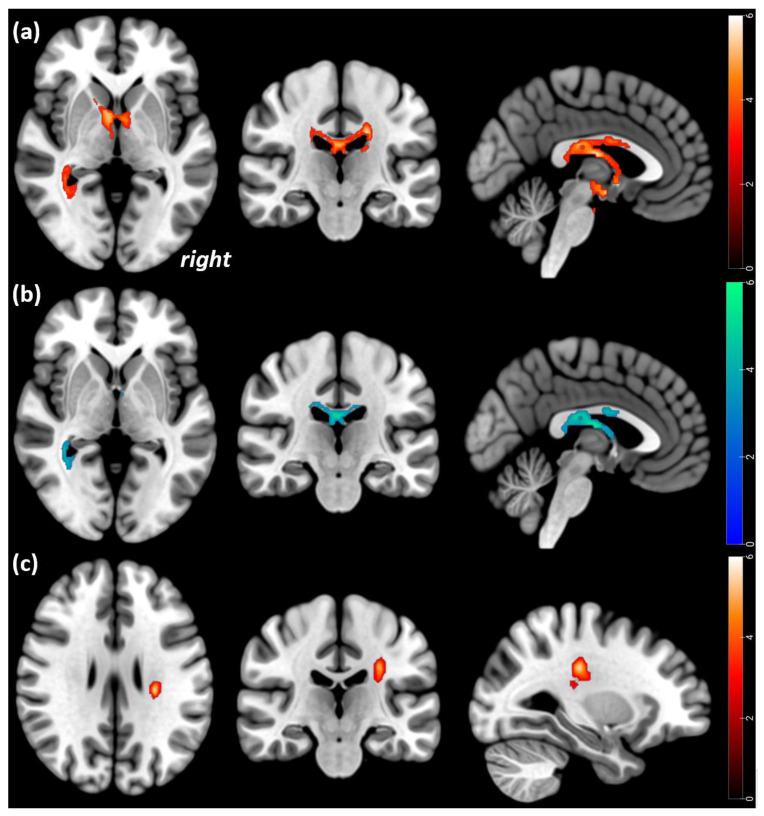
Statistical parametric mapping (t) intensity projection maps rendered onto a stereotactically normalized MRI scan: voxel cluster of the significant FA alterations in AN (statistical significance is thresholded at *p* < 0.001, FWE *p* < 0.05 corrected at the cluster level). (**a**) Lower FA in acute AN patients relative to HC at Tp1 (red color), (**b**) increases in AN patients from Tp1 to Tp2 (blue color), (**c**) decreases in AN patients from Tp1 to Tp2 (red color). The right side of the image corresponds to the right side of the brain.

**Figure 2 brainsci-13-00798-f002:**
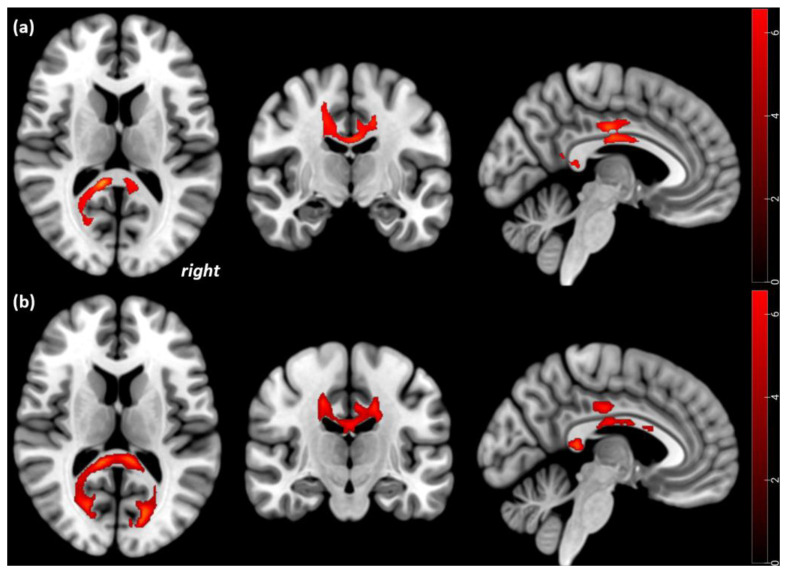
Statistical parametric mapping (t) intensity projection maps rendered onto a stereotactically normalized MRI scan: voxel cluster of the significant FA alterations in AN (statistical significance is thresholded at *p* < 0.001, FWE *p* < 0.05 corrected at the cluster level). FA decreases in acute AN patients with attachment trauma relative to HC at Tp1 (**a**) and at Tp2 (**b**). The right side of the image corresponds to the right side of the brain.

**Table 1 brainsci-13-00798-t001:** Description of our research study using PECO format.

Criteria	Description
Population	Adolescent inpatients between 14 and 18 years, with diagnosed anorexia nervosa (restrictive type) and attachment trauma
Clinical exposure	Standardized psychotherapeutic and nutritional treatment in patients with anorexia nervosa
Environmental exposure	Attachment trauma
Comparison	Adolescents between 14 and 18 years with no present or history of anorexia nervosa and no attachment trauma; adolescent inpatients between 14 and 18 years with diagnosed anorexia nervosa (restrictive type) and no attachment trauma
Outcome	White matter fiber microstructural alterations in anorexia nervosa assessed by diffusion tensor imaging

**Table 2 brainsci-13-00798-t002:** Age and clinical characteristics of the total AN sample, the resolved and the unresolved attachment groups and the healthy controls at timepoint 1 (baseline).

	acAN Cohort			HC Cohort
	Whole Cohort	Resolved Attachment	Unresolved Attachment	
Sample size	22	10	12	18
Age (years)	15.8 (1.2) *	16 (1.2) *	15.6 (1.3) *	17.7 (0.7) *
BMI Tp1 (kg/m^2^)	15.4 (1.4) *	15.5 (0.9) *	15.5 (1.4) *	21.2 (1) *°
BMI Tp2 (kg/m^2^)	17.8 (1) °	17.8 (1.2) °	17.7 (0.7) °	-
Duration of illness (months)	9.4 (6.8)	9.6 (4.8)	9.3 (8.3)	-
EDI-2 (total score)	298.6 (63) *	316 (44) *	293.4 (69) *	217.7 (58.4) *
Months between 1st and 2nd MRI scan	2.6 (0.9)	2.9 (1)	2.4 (0.8)	-
TIV (mm^3^)	1336.5 (114.1)	1366.6 (122.3)	1309.9 (116.8)	1369.8 (93.9)

acAN, acute anorexia nervosa patients at timepoint 1 (baseline); BMI, body mass index; EDI-2, Eating Disorder Inventory 2; TIV, total intracranial volume. Raw values are represented as mean (±1 standard deviation). The statistical tests are corrected for multiple comparisons (Holm–Sidak) at the 5% significance level. * Significant differences between AN and HC cohorts, ° BMI of AN patients at Tp2 compared to HC (*p* < 0.001) (no significant differences were found between the attachment subgroups). Note: Reprinted from [4]. Copyright 2023 by John Wiley & Sons.

**Table 3 brainsci-13-00798-t003:** Sociodemographic characteristics and attachment distributions between the patient and the healthy control groups.

	AN Group	HC Group	χ2	Φ	*p*
	*n* = 22 (%)	*n* = 18 (%)			
Number of siblings					
Single child	4 (18.2)	2 (11.8)	4.296	0.33	0.231
One sibling	9 (40.9)	6 (35.3)			
Two siblings	3 (13.6)	7 (41.2)			
More than two siblings	6 (27.3)	2 (11.8)			
Marital status of parents					
Married/partnership	9 (40.9)	11 (61.1)	1.616	−2.01	0.204
Single/divorced	13 (59.1)	7 (38.9)			
Occupation					
Attending school	21 (95.5)	17 (94.4)	0.021	0.02	0.884
Employed/trainee	1 (4.5)	1 (5.6)			
Attachment classifications					
Resolved/organized	10 (45.5)	14 (77.8)	4.310	−0.33	0.038
Unresolved/disorganized	12 (54.5)	4 (22.2)			

AN, anorexia nervosa; HC, healthy controls; resolved, adolescents with a secure, an insecure-dismissing or an insecure-preoccupied attachment pattern; unresolved/disorganized, adolescents with an attachment trauma. Level of significance *p* ≤ 0.05. Note. Reprinted from [4]. Copyright 2023 by John Wiley & Sons.

**Table 4 brainsci-13-00798-t004:** Longitudinal regional differences in FA values.

Significant regional differences in FA values in the 21 AN patients compared to the 18 HC participants							
Overlap of cluster region	kE	MNI coordinates			t value	*p*-value corrected at the cluster level (FWE)	Height Threshold
		x	y	z			
Significant FA decreases in the 21 acute AN patients compared to the 18 HC participants at Tp1							
Corpus callosum and cingulum bilaterally spreading to the fornix, and the corticospinal tract	1485	18 −20 14	−20 −36 0.34	26 4 9	6.4	<0.001	0.001
Inferior occipitofrontal and longitudinal fasciculus left	156	−32	−42	−4	5.6	0.029	
Significant FA increases in the 17 AN patients from Tp1 to Tp2							
Corpus callosum and cingulum bilaterally spreading to the fornix	885	2 −4 4	−12 −26 2	14 22 −1	6.7	<0.001	0.001
Inferior occipitofrontal and longitudinal fasciculus left	165	−34	−42	−4	4.5	0.008	
Significant FA decreases in the 17 AN patients from Tp1 to Tp2							
Corticospinal tract, right	125	26	−24	30	8.1	<0.001	0.001

**Table 5 brainsci-13-00798-t005:** Cross-sectional comparison for regional differences.

Significant regional differences in FA values in the 11 AN patients with attachment trauma compared to the 18 HC participants when corrected for BMI							
Overlap of cluster region	kE	MNI coordinates			t value	*p*-value corrected at the cluster level (FWE)	Height Threshold
		x	y	z			
Significant FA decreases in the 11 acute AN patients with attachment trauma compared to the 18 HC participants at Tp1							
Corpus callosum and cingulum bilaterally	3035	−10 14 16	−40 −6 −18	12 40 40	5.2	<0.001	0.01
Significant FA decreases in the 11 AN patients with attachment trauma compared to the 18 HC participants at Tp2							
Corpus callosum and cingulum, left	1393	−12	−40	12	4.4	0.005	0.01
Corpus callosum and cingulum, right	1312	14	12	38	3.8	0.007	

## Data Availability

The data that support the findings of this study are available upon request from the corresponding author. The data are not publicly available due to privacy or ethical restrictions.

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
