# Peer review of "Attachment Trauma Is Associated with White Matter Fiber Microstructural Alterations in Adolescents with Anorexia Nervosa before and after Exposure to Psychotherapeutic and Nutritional Treatment"

_brainsci, 2023, doi:10.3390/brainsci13050798_

Round 1

Reviewer 1 Report

Comments and Suggestions for Authors

The authors present an interesting topic attachment for microstructural white matter changes in adolescents with anorexia nervosa. From a medical point of view, the article is complete, but from a technical point of view, some clarifications are needed:

- for the statistical analysis, the SPM 12 application developed by a UK institute was used. I ask the authors to present in the article how the statistical analysis is carried out, which are the algorithms used for the voxel cluster, for example: 1) parametric statistics based on Random Field Theory); 2) nonparametric statistics based on permutation/randomization analyses; and 3) nonparametric statistics based on Threshold Free Cluster Enhancement.

- the SPM analysis method mentioned in the article was also used by other authors in articles reported in the specialized literature for similar medical cases. What is the novelty that the authors propose from the point of view of the proposed method?

Author Response

1) For the statistical analysis, the SPM 12 application developed by a UK institute was used. I ask the authors to present in the article how the statistical analysis is carried out, which are the algorithms used for the voxel cluster, for example:

1.1. parametric statistics based on Random Field Theory);

1.2. nonparametric statistics based on permutation/randomization analyses; and

1.3. nonparametric statistics based on Threshold Free Cluster Enhancement.

We thank the reviewer for these comments and added this information in the methods section accordingly.

Materials and methods (statistical analysis, page 13/14)

For statistical analyses of imaging data, a flexible factorial design was set-up to compare cross-sectional and longitudinal DTI data at baseline and follow-up time-point. A parametric model is assumed for each voxel that applies a general linear model to describe the data variability. Based on the number of resels in the image and the Random Field Theory adjustments were made to correct for spatial correlation and multiple comparisons. This was followed by correction for multiple comparisons via the family-wise error rate (FWE) at the p-level < 0.05 to adjust for type I errors (false positives) because numerous statistical tests are conducted.

2) The SPM analysis method mentioned in the article was also used by other authors in articles reported in the specialized literature for similar medical cases. What is the novelty that the authors propose from the point of view of the proposed method?

In our opinion, there is no simple answer regarding which of the numerous software packages (e.g. SPM, Freesurfer, TBSS, BrainVoyager, DoDTI, DTIstudio, Camino, etc.) is best or most suitable for this DTI analysis. We have chosen the well-established software package SPM12 to get a meaningful, voxel-based analysis and to detect focal DTI differences in the whole brain. A further reason for choosing this method is that we have a lot of experience with this software package. It was not the aim of this study to establish or expand a new method for imaging analysis. One advantage of the used approach is that there is no prior selection or segmentation of brain regions needed. This allows to investigate alterations in MRI parameters across the whole brain

Materials and methods (MRI acquisition and processing, page 13)

To pre-process and analyze DTI data, statistical parametric mapping (SPM, Wellcome Department of Cognitive Neurology, London, UK) was used. The software package SPM12 implemented in MATLAB 9.5 (R2018b; MathWorks, Natick, MA, USA) objectively localizes focal DTI differences throughout the whole brain at voxel-level (Friston et al.). One advantage of the used approach is that there is no prior selection or segmentation of brain regions needed. This allows to investigate alterations in MRI parameters across the whole brain.

Reviewer 2 Report

Comments and Suggestions for Authors

Thank you for giving me the opportunity to review this manuscript.

I think it is necessary to revise it.

1) Please indicate the study's design more clearly in the title, the abstract and the method section. According to the ClinicalTrials.gov, this study is a case-control study. If so, please describe it clearly. Furthermore, please describe it by using the PECO (the population, the exposure, the control, and the outcomes) format. in the method section. Furthermore, please describe why both cross-sectional and longitudinal comparisons were performed because I cannot understand it. 

2) Please describe the relevant dates, including periods of recruitment, exposure, follow-up and data collection.

3) Please describe the sources and methods of case ascertainment and control section. Please describe the rationale for the choice of cases and controls. Please describe matching criteria.

4) Please describe any potential confounders and effect modifiers. Please describe any statistical methods including those used to control for confounders. Please describe unadjusted estimates and confounder-adjusted estimates and their precision, and make clear which confounders were adjusted and why they were included.

5) Please describe how sample size was arrived at.

6) Please describe how missing data were addressed. Please describe number of participants with missing data for each variable of interest.

7) In this study, patients received psychological therapy. why is it possible to investigate the association between attachment trauma and microstructure in patients with AN? Furthermore, were white matter abnormalities evident in AD compared with HC? Is it possible to demonstrate the association between attachment and WM abnormalities in this study design?

8) The authors described in the introduction that "these results have been relatively inconsistent and may arise from differences in subject characteristics (e.g., duration of illness), methodological approaches and neurodevelopmental factors." Please describe in detail how to overcome these issues in this study by improving its study design.

It is necessary to revise the manuscript.

I think that the study design was disorganized.  

Author Response

1) Please indicate the study's design more clearly in the title, the abstract and the method section. According to the ClinicalTrials.gov, this study is a case-control study. If so, please describe it clearly. Furthermore, please describe it by using the PECO (the population, the exposure, the control, and the outcomes) format. in the method section. Furthermore, please describe why both cross-sectional and longitudinal comparisons were performed because I cannot understand it.

And 2) Please describe the relevant dates, including periods of recruitment, exposure, follow-up and data collection.

And 3) Please describe the sources and methods of case ascertainment and control section. Please describe the rationale for the choice of cases and controls. Please describe matching criteria.

In regard to comment 1, we indicate the study design more clearly in the title by changing it to “attachment trauma is associated with white matter microstructure in adolescents with anorexia nervosa before and after exposure to psychotherapeutic and nutritional treatment”. Furthermore, we defined the study design more precisely according to PECO in the abstract and the method section (see participants section). We added a table presenting our research question according to PECO format. In regard to comment 2, we described relevant dates in the participants section. Concerning comment 3, we added the following information: cases and controls were ascertained by using the ICD-10 F50.0 criteria for Anorexia Nervosa. We assessed these with the SCID-1 interview, a reliable clinical instrument for adults and adolescents. They were gender-matched (see participants section). In-text changes were as follows:

Introduction (hypothesis, page 7)

…the hypothesis of the second part of the study was that the AN patients with unresolved attachment status show persisting WM fiber microstructural alterations as assessed by DTI,…

Materials and methods (participants, page 8, table 1)

Our case population consisted of 27 female adolescent inpatients from the Department of Child and Adolescent Psychiatry with a diagnosed Anorexia Nervosa (ICD-10 F50.0) and with or without attachment trauma. We recruited a gender-matched control population of 23 non-clinical adolescents from the community with no present or history of Anorexia Nervosa (ICD-10 F50.0) and no attachment trauma…The recruitment took place between January 2015 and June 2018. The cases and controls were ascertained by using the ICD-10 F50.0 criteria for Anorexia Nervosa restrictive type (present-absent). Diagnoses were based on the Structured Clinical Interview for DSM-IV (SCID-I [47])… The case sample was exposed to psychotherapeutic treatment that included…MRI scans at Tp1 and Tp2, inpatient treatment and data collection were done between February 2015 and September 2018. We present our research question in a population, exposure, comparison and outcome format (PECO) in table 1.

4) Please describe any potential confounders and effect modifiers. Please describe any statistical methods including those used to control for confounders. Please describe unadjusted estimates and confounder-adjusted estimates and their precision, and make clear which confounders were adjusted and why they were included.

We agree with the reviewer and add this information. Age and TIV are strongly recommended to use as covariates for all VBM analyses to correct for different brain sizes. Therefore, they were included in all analyses as covariates. BMI, which was strongly associated with recovery of gray and white matter regions, was included as nuisance variable in group analyses regarding attachment trauma (2nd study part). We did that to correct for weight- and nutrition-associated effects and to reduce the associated variance, because the aim of this analysis was to detect attachment status related differences.

Materials and methods (statistical analysis, page 14)

Total intracranial volume and age were used as nuisance variables in all analyses to correct for different brain sizes. In the second part of the study, BMI was also included as a covariate in statistical models to minimize the effect of weight- and nutrition-associated effects, as the goal of this study part was to detect attachment status related differences in DTI metrics.

5) Please describe how sample size was arrived at.

Our sample size is comparable to previous studies exploring WM alterations in Anorexia Nervosa. As some of our patients and controls dropped out of the study (see exclusion criteria in the materials and methods section, participants), we arrived at a final sample size of N = 22 AN patients and N = 18 controls. Of course, higher sample sizes would be preferable and would allow further conclusions on WM microstructure in adolescent AN. We added that aspect in the limitation section.

Discussion (page 25)

Fifth, although our sample size is comparable to those in previous brain imaging studies in AN [61], our results need to be replicated in large scale research.

6) Please describe how missing data were addressed. Please describe number of participants with missing data for each variable of interest.

We added this information.

Materials and Methods (participants, page 9)

There were no missing data in the final sample.

7) In this study, patients received psychological therapy. Why is it possible to investigate the association between attachment trauma and microstructure in patients with AN? Furthermore, were white matter abnormalities evident in AD compared with HC? Is it possible to demonstrate the association between attachment and WM abnormalities in this study design?

In our study design patients received, beside nutritional therapy, psychotherapy in a standardized, non-individualized scheme referring to a variety of treatments that aim to help a person identify and change negative emotions, thoughts, and behaviors. They did not receive attachment-related individualized therapy, which retrospectively might be an improvement in therapy in the view of our results and represents an interesting aspect for future research. Attachment trauma is a population-based factor, which can be used to divide the collective into two subgroups and to compare them cross-sectionally to healthy controls and in the longitudinal course.

Yes, we found DTI differences in AN patients with attachment trauma compared to controls. We provided that information in the results section, please see “cross-sectional comparisons of patients with attachment trauma” and figure 2/ table 5. No significant differences were found in AN individuals without attachment trauma compared to healthy controls when corrected for BMI. We also found significant differences in cross-sectional comparisons of all AN patients compared to healthy controls, which is outlined in the chapter “cross-sectional comparisons of FA he whole group comp”.

There are two main theories that consider DTI alterations in AN patients as (1) reflections of predisposing personality traits, and as (2) effects of undernutrition (Monzon et al., 2016). Other research papers have previously demonstrated that WM microstructure changes from pre- to post therapy (see for example Griffiths et al., Schwanenflug et al., etc.). However, mechanisms underlying these changes have only recently been investigated (see for example Lenhart et al., 2022 who demonstrated that attachment trauma is associated with gray matter alterations from pre- to post therapy; or Schwanenflug et al., 2018 who explored psychological components like depression and anxiety of gray and white matter alterations from pre- to post therapy). Based on these recently published findings in the field of gray matter, we investigated a similar question in relation to WM microstructure as attachment trauma might also be related to WM alterations. In other words, patients with attachment trauma might be less responding to traditional psychotherapeutic treatment and thus one could hypothesize that a treatment focusing on attachment trauma might be of particular importance for these patients - a hypothesis which of course has to be tested in future studies. Furthermore, our approach provides a more differentiated perspective on the inconsistent findings reported in other studies by including psychological factors.

We outlined these aspects in the paper in the introduction (page 7).

Although, some research groups have previously demonstrated that WM microstructure changes from pre- to post therapy ([5,22,23], research on underlying psychological factors associated with these changes has only recently begun. Whereas psychological components like symptoms of anxiety or depression were not related to gray and white matter changes in AN patients [5], Lenhart et al. [4] demonstrated in a recently published study that attachment trauma is associated with gray matter alterations pre-post therapy. Given that WM alterations follow a similar pattern [5], attachment might also play a key role for changes from pre- to post treatment. Therefore, we hypothesized that attachment trauma might also be related to WM alterations in AN patients suggesting they might be less responding to traditional psychotherapeutic and nutritional treatment.

8) The authors described in the introduction that "these results have been relatively inconsistent and may arise from differences in subject characteristics (e.g., duration of illness), methodological approaches and neurodevelopmental factors." Please describe in detail how to overcome these issues in this study by improving its study design.

In the present study, we circumvent some of these aspects  by (1) focusing exclusively on an adolescent age group between 14 and 18 years of age, (2) methodologically we included attachment trauma as another component that might be important to explain the observed WM alterations pre- and post therapy, (3) as our population is of a young age we do not have any chronic forms of AN with several years of illness in the past that might demonstrate poorer therapeutic response and thus show other WM alterations after treatment. In addition, our drop-out rate for Tp2 was relatively small so our sample size for a longitudinal design was relatively high compared to other studies (i.e. Travis et al., Griffiths et al.) which increases the power of our results. We added that information in the final paragraph of the discussion section.

We added that information in the final paragraph of the discussion (page 25/26)

Despite these limitations, the present study has a number of strengths as it circumvents some aspects that might be responsible for the inconsistent results reported in previous longitudinal studies on WM microstructure in patients with AN. On a neurodevelopmental level, our research focused exclusively on an adolescent age group between 14 and 18 years of age. As our population is of a young age we do not have any chronic forms of AN with several years of illness in the past that might demonstrate poorer therapeutic response and thus show different WM alterations after treatment. Methodologically, we included attachment trauma as a component that might be important to explain the observed WM alterations pre- and post-therapy. In addition, our drop-out rate at Tp2 was relatively small so our sample size for a longitudinal design was relatively high compared to other studies, (i.e. [Travis et al., Griffiths et al.]).

Reviewer 3 Report

Comments and Suggestions for Authors

The manuscript reports an interesting longitudinal study in AN, looking at the possible effects of traumatic events on brain white structure. The paper has an interesting approach to a hot topic, which is the biological effects of trauma in the ED population. The paper is well-written and the methods are reported clearly. The authors have correctly reported the limitations of the results (like the small sample size).

I have some comments that require the evaluation of the authors:

- a recent meta-analysis of the FA data in AN (https://doi.org/10.1002/eat.23160) reported similar data to this study but concluded by calling for an evaluation of free-water correction in the analysis. Have you evaluated this aspect? 

- traumatic events have been shown to reduce the daily production of cortisol with a possible central effect, with the proposal of an ecophenotype in ED patients (https://doi.org/10.1002/erv.2896). Your data seems to be coherent with this evidence. I think you might discuss this aspect.  

- It is not clear the modality used to assess the presence of trauma. Have you excluded participants with different traumatic histories?

- Who performed the AAP scoring? Have you used a consensus? 

- You deleted the phrase about the evaluation of the control group. Is there a reason? Were controls evaluated?

- Did you wait for a stable weight recovery? How was it defined?

Author Response

Comment 1: a recent meta-analysis of the FA data in AN (https://doi.org/10.1002/eat.23160) reported similar data to this study but concluded by calling for an evaluation of free-water correction in the analysis. Have you evaluated this aspect? 

Answer: We thank the reviewer for mentioning this noteworthy aspect. In an analysis, focusing on the fornix Kaufmann et al. demonstrated that DTI metrics may be biased by CSF-induced partial volume effects (PVE) due to ventricular enlargement typically found in acute AN. Although they did not analyze the whole brain, this data suggest that other periventricular regions could also be affected by PVE. We addressed this issue by including ventricular size as a covariate in our general linear model. Consistent with previous studies (von Schwanenflug et al., 2018), ventricular volumes were negatively correlated with FA values predominantly of the fornix, but also other periventricular regions were significantly associated with FA alterations in acute AN (please see supplemental table and figure). Small parts of the corpus callosum might be also affected by PVE, although the changes were not extensive. We also added ventricular size as a covariate in the analyses focusing on AN patients with attachment trauma, but did not find any additional influences of ventricular size on FA metrics. Because of the extensive shrinkage of GM and WM in acute AN, the introduction of a covariate like BMI, which was already included as a covariate in attachment related analyses, may also partially correct for free water itself. We agree that free water correction may be a protentional interesting factor for future studies.

We added these paragraph in the manuscript (discussion section/ limitations), and following additions in methods and results section.

Methods, statistical analysis:

Additionally, ventricular size was included as a covariate in our general linear model the test the potential influence of ventricular size on DTI metrics (Kaufmann et al., 2017). 

Results, cross-sectional comparisons of FA:

Ventricular size was negatively correlated with FA values predominantly of the fornix and adjacent periventricular regions (P < 0.001), and the inferior occipitofrontal and longitudinal fasciculus on the left side (P = 0.003) (see supplemental table and figure).

Results, cross sectional comparisons of patients with attachment trauma:

Further, no associations between ventricular size and DTI metrics were evident.

Discussion: Concerning methodology, a recent meta-analysis reported similar data to our study and called for an evaluation of free-water correction in the analysis [66]. In an analysis focusing on the fornix Kaufmann et al. [67] demonstrated that DTI metrics may be biased by CSF-induced partial volume effects (PVE) due to ventricular enlargement typically found in acute AN suggesting that the fornix and other periventricular regions might be affected by PVE. We addressed this issue by including ventricular size as a covariate in our general linear model. Consistent with previous studies ([5], ventricular size was negatively correlated with FA values predominantly of the fornix, but also other periventricular regions were significantly associated with FA alterations in acute AN. Small parts of the corpus callosum might be also affected by PVE, although the changes were not extensive. Results of attachment related analyses were not influenced from adding ventricular size as a covariate in these analyses. Because of the extensive shrinkage of GM and WM in acute AN, the introduction of a covariate like BMI, which was already included as a covariate in the attachment-related analyses, may also partially correct for free water itself. Additionally, the lack of evaluation of free-water correction might be related to a reduction in effect size so that our findings might underestimate the observed association. Nevertheless, free water correction may be a protentional interesting factor for future studies.

Comment 2: traumatic events have been shown to reduce the daily production of cortisol with a possible central effect, with the proposal of an ecophenotype in ED patients (https://doi.org/10.1002/erv.2896). Your data seems to be coherent with this evidence. I think you might discuss this aspect.  

Answer: We thank the reviewer for this useful and interesting comment. We added this aspect in our discussion section.

Discussion section: Furthermore, research studies suggest an impact of childhood trauma on biological traits like a heightened sensitivity to salient stimuli that results in inappropriate stress responses. For example, the study of Meneguzzo et al. (2021) tested 24-h urinary free cortisol levels during the acute phase in patients with eating disorders (AN or Bulimia Nervosa). They found a reduced daily excretion of cortisol in patients with a history of childhood maltreatment compared to those with no maltreatment experiences. In line with the observed impaired functioning of the HPA axis found in the Meneguzzo et al. study (2021), our results might add further data supporting the hypothesis of a presence of an ecophenotype of maltreated AN. Future research incorporating this phenotype into the evaluation of personalized treatment approaches might advance our understanding of the clinical presentation, outcome and treatment efficacy in the field of AN.”

Comment 3: It is not clear the modality used to assess the presence of trauma. Have you excluded participants with different traumatic histories?

Answer: Our study focuses on attachment trauma in particular. This form of trauma is defined as a state of dysregulation and helplessness because these individuals cannot find solace and comfort when facing severe stressors. Attachment trauma is rooted in past experiences of threatened abandonment (i.e. parental loss) or danger by an attachment figure (i.e. abuse) and can be assessed using attachment interviews like the Adult Attachment Projective Picture System or the Adult Attachment Interview. We added that information in the introduction section. We did not assess or analyze associations of other forms of trauma (i.e. exposure to or being the witness of terrible events like war, accidents, natural disasters etc.). It was not an exclusion criterion to suffer from another form of trauma as patients with an attachment trauma can also have experienced these. We outlined this aspect in the discussion/limitation section.

Introduction section: Attachment trauma is considered as a form of relational trauma and its presence can be assessed using an attachment interview (i.e. the Adult Attachment Projective Picture System (AAP) or the Adult Attachment Interview (AAI)).

Discussion section: Furthermore, we exclusively focused on attachment trauma and we did not consider other forms of trauma (i.e. exposure to or being the witness of terrible events like war, serious accidents, natural disasters etc.) in our study. We did not exclude participants who suffered from another form of trauma as patients with attachment trauma can also have experienced these. Research even demonstrated associations between attachment trauma and a higher level of PTSD symptoms after traumatic events (Gander et al., 2018). Thus, more studies with larger sample sizes are needed to explore WM abnormalities in individuals with other forms of trauma.

Comment 4: Who performed the AAP scoring? Have you used a consensus? 

Answer: AAP scoring was done by 2 independent certified AAP judges. To become an AAP judge you need to participate in an intensive workshop and reliably code 80% of a minimum of 30 AAP cases (for certification guidelines see George & West, 2011). We added that information in the methods section. Furthermore, we added data on interrater reliability. Disagreements were resolved by conference.

Methods section: The AAP interviews were coded by two certified and reliable AAP judges (AB and MG,), who participated in an 8-day AAP workshop and reliably coded 80% on a minimum of 30 cases (for more information on certification criteria see George & West, 2011). One of the judges was a member of the International AAP Training Consortium and led several neurophysiological and clinical research studies since 2002. Inter-rater reliability for this study demonstrated an agreement for the two-group classification (resolved-unresolved), it was 97.5%, κ = 0.947 with a narrow 95% confidence interval [0.845, 1.049], p < 0.001. The independent judges agreed in 39 out of 40 cases and disagreements were resolved by conference.

Comment 5: You deleted the phrase about the evaluation of the control group. Is there a reason? Were controls evaluated?

Answer: We rephrased that passage in response to a reviewer’s comment (“the cases and controls were ascertained by using the ICD-10…”). Yes, we evaluated controls and therefore we decided to add the following sentence:

Participants section: “The control group was assessed with the SCID-I interview to exclude all participants with a present or a history of a psychiatric disorder (i.e. eating disorders, anxiety disorders, major depression).”

Comment 6: Did you wait for a stable weight recovery? How was it defined?

Answer: Yes, Tp2 was done after stable weight recovery. According to WHO and CDC this is defined by a BMI < 18.5 kg/m². As BMI-for-age ≤ 5th percentile is commonly used for children and adolescents (Engelhardt et al., 2021), we used that definition for our study. We added that information in the methods section.

Methods section: In the ICD-11 AN is characterized by a significantly low body weight for the person’s height, developmental stage and age that is not due to the unavailability of food or other medical conditions. In adults, the diagnosis of AN depends on a BMI < than 18.5 kg/m² as defined by the Centers for Disease Control and Prevention (CDC) and the World Health Organization (WHO). A BMI-for-age ≤ 5th percentile is commonly used for children and adolescents (Engelhardt et al., 2021). For our study, a stable weight recovery was therefore defined by reaching a BMI-for-age ≤ 5th percentile.

Round 2

Reviewer 2 Report

Comments and Suggestions for Authors

The study design was not organized at all. It is impossible to assess any relationship by this design. Different populations were included in the same PECO.

Author Response

Answer: We apologize that the study design still seems to be disorganized. Due to the empirical nature of our manuscript, we are afraid that the PECO format, which is almost exclusively used for reviews, does not add a significant value in comparison to the revised version of the manuscript. Nevertheless, we want to use the opportunity to explicitly state the conceptual approach reported in the present paper. The environmental exposure variable was attachment trauma classified by the AAP, which was performed with all participants enrolled in this study. The clinical exposure variable was the nutritional and psychotherapeutic treatment, which was done with the case sample (N=22). The outcome variable were white matter fiber microstructural alterations using DTI, which we investigated in two subgroups (patients with attachment trauma vs. patients with no attachment trauma) before and after treatment. We added the following paragraph in the methods section:

Methods section: The present study has a pre-post intervention design with attachment trauma (classified by the AAP) as the environmental exposure and nutritional and psychotherapeutic treatment as the clinical exposure. The outcome variable were white matter microstructural alterations assessed by DTI. In addition, WM alterations were compared between patients and controls (Tp1) and between patients with attachment trauma and patients with no attachment trauma (Tp1 and Tp2).

Reviewer 3 Report

Comments and Suggestions for Authors

I think the authors have addressed all my concerns and seriously improved the manuscript. Thus I recommend acceptance. 

Author Response

The authors want to thank the reviewer for the time and invaluable suggestions  to improve our manuscript. We made the required spell check.